# Unraveling the Pathophysiology of Irritable Bowel Syndrome: Mechanisms and Insights

**DOI:** 10.3390/ijms262110598

**Published:** 2025-10-30

**Authors:** Ioanna Aggeletopoulou, Konstantinos Papantoniou, Ploutarchos Pastras, Christos Triantos

**Affiliations:** Division of Gastroenterology, Department of Internal Medicine, University of Patras, 26504 Patras, Greeceploutarchosp96@gmail.com (P.P.)

**Keywords:** irritable bowel syndrome, pathophysiology, mechanistic insights, gut microbiome, brain–gut axis, serotonin metabolism, bile acids, gut barrier integrity

## Abstract

Irritable Bowel Syndrome (IBS) is one of the most prevalent gastrointestinal disorders, affecting about 11% of the global population and exerting a substantial burden on quality of life and healthcare systems. Despite the emerging interest in this disease, its pathophysiology remains elusive, reflecting the interplay between the brain–gut axis, neuroendocrine dysregulation, immune activation, barrier dysfunction, microbial imbalance, and environmental triggers. Disruptions in the hypothalamic–pituitary–adrenal axis, impaired serotonin signaling, bile acid malabsorption, and altered intestinal permeability collectively result in the emergence of abnormal motility, visceral hypersensitivity, and chronic inflammation. The gut microbiome further modulates these processes by influencing neurotransmitter metabolism, immune responses, and epithelial integrity, positioning it as both a driver of symptoms and a promising therapeutic target. The aim of this review is to synthesize current mechanistic insights into IBS, highlighting the interconnected roles of the brain–gut axis, immune modulation, and microbial dynamics, and to explore how these pathways may be translated into precision medicine approaches. This review integrates molecular, microbial, and neuroimmune perspectives to propose a systems-level understanding of IBS pathophysiology and its implications for precision medicine. By integrating host–microbe interactions, dietary influences, and genetic predispositions, we highlight the mechanistic complexity underlying IBS and the potential for translating these insights into personalized strategies for symptom control and improved quality of life.

## 1. Introduction

Irritable bowel syndrome (IBS) is a complex functional gastrointestinal disorder characterized by chronic or recurrent abdominal pain, stool irregularities, and bloating [1,2]. The global prevalence of IBS is estimated at about 11%, with considerable variation across geographic regions. The condition significantly affects patients’ quality of life [1] and is generally considered female-predominant [3]. The IBS diagnosis is challenging due to its fluctuating symptoms and their similarity to those of other gastrointestinal disorders [4,5]. IBS is classified into four subtypes based on symptoms, including the constipation-predominant IBS (IBS-C), diarrhea-predominant IBS (IBS-D), mixed-type IBS (IBS-M), and unclassified IBS (IBS-U), and is often associated with somatic comorbidities, psychiatric conditions, and visceral sensitivity [1,6]. Due to the lack of a definitive diagnostic test for IBS, symptom-based criteria have been developed through expert consensus to improve diagnostic consistency and minimize unnecessary testing. The most recent update, Rome IV (2016), introduced stricter criteria by removing abdominal discomfort and augmenting the pain frequency threshold to at least once per week [7]. As a result, fewer people are diagnosed with IBS under the Rome IV criteria, but those who are diagnosed tend to have more severe symptoms and higher levels of psychological distress [8]. First-line management of IBS typically involves dietary modification, the use of soluble fiber, and antispasmodic agents. For patients with more severe or persistent symptoms, additional options include central neuromodulators, intestinal secretagogues, agents targeting peripheral opioid or 5-hydroxytryptamine (5-HT) receptors, and antibiotics with predominant intraluminal activity. Non-pharmacological strategies, including mind–body interventions and psychotherapy, also play an important role in symptom control [6].

Although IBS is widely recognized as a multifactorial condition, its underlying mechanisms remain poorly understood. More recently, IBS and related functional gastrointestinal disorders have been reclassified as disorders of gut–brain interaction (DGBI), characterized by gastrointestinal symptoms that may result from various combinations of motility abnormalities, visceral hypersensitivity, immune and mucosal dysfunction, gut microbiota alterations, and disturbances in central nervous system (CNS) processing [2,9]. Emerging evidence also supports its developmental role. Early-life stress, infections, and nutritional factors can disrupt microbiota maturation and neuroendocrine programming, leading to long-term alterations in hypothalamic–pituitary–adrenal (HPA) axis activity, immune tolerance, and visceral sensitivity [10,11]. These early disturbances may predispose individuals to increased stress reactivity and altered gut–brain communication, thereby increasing susceptibility to IBS and other disorders of gut–brain interaction.

Emotional and psychological stressors, as well as food intolerances, further shape the clinical presentation and can worsen symptom severity [12,13]. Genetic susceptibility and epigenetic mechanisms have also been implicated, although their contributions appear modest compared to environmental influences [14]. Finally, the gut microbiota contributes to IBS pathophysiology by modulating intestinal gene expression and host physiological responses through its metabolites [15,16].

The aim of the current review is to synthesize current mechanistic insights into IBS, emphasizing the interplay between the brain–gut axis, immune modulation, and gut microbiota. By bringing together current evidence across multiple systems, we underscore the need for a multidimensional understanding of IBS as a chronic disorder influenced by interconnected physiological, psychological, and microbial parameters.

## 2. Materials and Methods

A comprehensive literature search was performed in the PubMed, Medline and Scopus databases for articles published up to September 2025 using combinations of the terms “Irritable bowel syndrome” or “IBS” and “pathophysiology”, “microbiome”, “serotonin”, and “immune modulation”, “brain–gut axis”,” bile acids,” molecular mechanisms” and “gut barrier integrity”. Relevant original and review articles were selected to illustrate the mechanistic and translational aspects of adult IBS. Evidence from experimental or animal studies was included only when it provided mechanistic insight relevant to human disease.

## 3. The Brain–Gut Axis in IBS: From Neuroendocrine Regulation to Clinical Manifestations

IBS is considered the result of dynamic interactions between genetic, environmental, and physiological factors, including dysregulation of the autonomic nervous system (ANS) and other gastrointestinal influences [17,18]. A key aspect is the bidirectional interplay between the CNS and the enteric nervous system (ENS), which together form the brain–gut axis, which serves as a central mechanism in IBS pathophysiology [18,19]. This bidirectionality allows the brain to influence gut function and the gut to impact brain processes, thereby disrupting gastrointestinal motility, visceral sensitivity, and immune regulation [20]. Disruption of this axis is considered a primary pathogenic mechanism in DGBI [21]. Communication between the CNS and ENS occurs through intricate neuronal, endocrine, immune, and metabolic signaling pathways that can be shaped by genetic predisposition, dietary factors, stress, physical activity, cognition, and social interactions [22]. The gut microbiota also plays a pivotal role, producing a broad array of neurotransmitters that regulate ENS activity locally, while growing evidence indicates indirect effects on CNS function [15,20]. *Bifidobacteria* can increase tryptophan availability, the precursor of serotonin [23], whereas *Lactobacillus* species influence γ-aminobutyric acid (GABA) metabolism and GABA receptor expression in the brain [24]. Although certain neurotransmitters produced in the gut can cross the intestinal mucosa and enter the circulation, they generally cannot pass through the blood–brain barrier, indicating that their effects on the brain are indirect [25]. Another important pathway linking the gut and the brain is the activation of the HPA axis, whereby psychological or physical stress triggers cortisol release, leading to alterations in gut microbiota composition and barrier integrity [26].

## 4. HPA Axis Dysregulation in IBS Pathophysiology

Dysregulation of the HPA axis is of particular importance in IBS, as elevated cortisol and stress-related mediators such as interleukin 6 (IL-6) and IL-8 have been consistently observed [27,28]. These impairments not only exacerbate gastrointestinal dysfunction but also predispose individuals to psychiatric comorbidities, thereby creating a vicious cycle between gut disturbances and mental health disorders [29,30]. Notably, IBS often precedes psychiatric manifestations, suggesting that gut dysfunction may contribute to psychological disturbances through various mechanisms, such as altered neurotransmitter metabolism [31,32]. This bidirectional interaction highlights the need for treatment approaches targeting both the gut and the brain. Emerging evidence, including findings of increased serine protease activity in IBS-D stool samples [33] and the reproduction of gut dysfunction in rodent models following microbiota transplantation [34], underscores the importance of integrated therapeutic strategies that address both neuroendocrine dysregulation and microbial imbalance. The downstream consequences of neuroendocrine dysregulation extend to mucosal immune activation and barrier integrity, linking central stress responses with peripheral gut alterations.

## 5. Mucosal and Systemic Immune Dysregulation: Linking Inflammation to Symptoms

In addition to neuroendocrine mechanisms, local and systemic immune responses represent another interface through which stress and microbial factors shape IBS pathophysiology. An acute gastrointestinal infection may initiate prolonged mucosal and systemic inflammation in susceptible individuals, thereby sustaining symptoms. Epidemiological data support this concept, showing that individuals with a history of enteric infection more frequently report IBS-like symptoms compared to unexposed individuals [35,36,37]. Immunohistological evidence has consistently demonstrated inflammatory changes in the intestinal mucosa of patients with DGBIs, particularly those with post-infectious IBS and IBS-D [38]. These include lymphocytic infiltration of both the small intestine and colon [39], with jejunal biopsies revealing increased lymphocyte infiltration of the myenteric plexus in 90% of patients and secondary neuronal degeneration in 60% [40]. These alterations are thought to arise from the release of mediators such as proteases, nitric oxide, and histamine by lymphocytes, which can activate the ENS and induce abnormal motor responses. Histological findings further support this inflammatory environment, including colonic eosinophilia and spirochetosis, as well as increased mucosal mast cells in close proximity to enteric nerves [41]. Mast cells release histamine, proteases, prostaglandins, and cytokines that sensitize nociceptors, contributing to visceral hypersensitivity and barrier dysfunction [42]. Recent evidence has also identified corticotropin-releasing factor (CRF) within eosinophils of the intestinal mucosa, showing higher CRF expression and enhanced eosinophil degranulation in patients with IBS-D compared to healthy controls [43]. These changes correlated with symptom severity, stress, and depression [43].

Beyond localized mucosal changes, systemic immune activation has been demonstrated. Elevated levels of pro-inflammatory cytokines have been reported in colonic mucosa and peripheral blood mononuclear cells, particularly in IBS-D patients, with higher concentrations correlating with anxiety and depression [39,44,45,46]. Humoral immune activation has also been described, with microarray analyses showing enhanced mucosal B-cell proliferation and immunoglobulin production in IBS-D, which correlated with stool characteristics and psychological comorbidity [47,48]. Functional data further support an immune contribution, as supernatants from immune cells of IBS-D patients enhanced nociceptive signaling in murine colonic afferents via cytokines, such as tumor necrosis factor alpha (TNFα), whereas this effect was absent in IBS-C supernatants [49]. In addition, reduced β-endorphin release from immune cells has been associated with impaired endogenous analgesia, providing another mechanism for visceral hypersensitivity [50].

## 6. Barrier Dysfunction and Epithelial Integrity: Molecular Mechanisms and Clinical Implications

Another key mechanism involves impairment of the intestinal epithelial barrier. A great number of patients with IBS, particularly those with IBS-D, present impaired intestinal permeability [51]. Disruption of barrier function has been linked to mast cell activation, dietary antigens, microbial alterations, and mediators, such as serotonin and proteases [51,52]. These changes correlate with pain severity and stool abnormalities [53,54]. At the molecular level, down-regulation of the TESK1/CFL (testis-associated actin remodeling kinase/cofilin 1) pathway has been identified in the jejunum of women with IBS-D, a pathway crucial for cytoskeletal remodeling and, secondarily, intestinal motility [54]. Recent evidence indicates that increased intestinal permeability is not limited to IBS-D and post-infectious IBS, but is also present in IBS-C (4–25% of patients), where it has been linked to mast cell activation and enhanced visceral hypersensitivity [42]. Activated mast cells adjacent to sensory nerve endings, along with increased TRPV1-expressing sensory fibers, have been directly linked to abdominal pain severity across IBS subtypes [55].

These mechanistic insights have therapeutic implications. The NHE3 inhibitor tenapanor has shown promise in restoring epithelial integrity and reducing pain in IBS-C [56]. Mucoprotectants, such as xyloglucan, gelatin tannate, and pea protein–tannin complexes, are emerging therapeutic agents aiming at restoring barrier integrity and reducing mucosal inflammation in IBS-D, with early clinical trials reporting improvements in stool consistency and abdominal symptoms [52]. A recent bibliometric analysis highlights the growing global interest in intestinal permeability in IBS, diet and gut microbiota interactions, underscoring nutrition-based strategies as promising avenues for modulating barrier function and symptoms [57].

Consistent with these findings, zonulin, a regulator of tight junctions, has been reported to be elevated in IBS-D, supporting compromised barrier integrity [58]. This disruption is further reflected in altered expression of tight junction proteins, such as occludin and claudin-1, and increased epithelial gaps [59]. In patients with suspected food intolerance, exposure to dietary antigens rapidly induces lymphocyte infiltration, epithelial disruption, and intervillous space widening, consistent with impaired barrier function [60]. Stress further amplifies this process through activation of the HPA axis, with cortisol release promoting mucosal immune activation and increasing pro-inflammatory cytokines [61].

Collectively, these immune and barrier disturbances support a disease model in which low-grade inflammation, immune dysregulation, mast-cell activation, and epithelial barrier dysfunction drive symptom development, particularly in IBS-D and post-infectious IBS. Although it remains uncertain whether these alterations represent causal agents or secondary consequences, their persistence highlights the need for standardized, longitudinal studies to clarify temporal relationships. Importantly, barrier disruption facilitates antigen exposure and immune sensitization, which may, in turn, influence neurotransmitters and bile acid signaling. These interconnected pathways help translate molecular and immune alterations into the diverse clinical manifestations of IBS.

## 7. Serotonin and Bile Acid Dysregulation: Converging Pathways in IBS Pathophysiology

Dysregulation of the brain–gut axis affects multiple gastrointestinal functions such as the sensory, motor, autonomic, and secretory processes [62]. These disturbances contribute to alterations in intestinal motility, permeability, visceral hypersensitivity, and gut microbiota disturbance. Serotonin (5-HT) metabolism has emerged as a key player in IBS, influencing gut physiology by regulating peristalsis, secretion, and pain perception [63]. Nearly 90% of the body’s serotonin is stored in enterochromaffin cells of the gut, where it functions as a sensor of luminal stimuli [64,65]. Once released, serotonin acts on intrinsic and extrinsic afferent neurons, orchestrating gastrointestinal motility and signaling to the central nervous system [66,67]. The effect of serotonin is limited by its reuptake through the serotonin transporter (SERT) in enterocytes, followed by metabolism to 5-hydroxyindoleacetic acid (5-HIAA) [68]. Impaired modulation in serotonin release, whether in the form of excess or deficiency, contributes to the hallmark symptoms of IBS diarrhea and constipation, respectively [69].

Evidence from patients with post-infectious IBS revealed persistently increased numbers of enterochromaffin cells compared to individuals who recovered from acute infection, suggesting long-lasting remodeling of serotonin signaling [70,71]. Meal-challenge studies subsequently demonstrated exaggerated postprandial 5-HT release in IBS-D, reduced release in IBS-C, and distinct alterations in the 5-HIAA/5-HT ratio, consistent with impaired reuptake in IBS-D and insufficient release in IBS-C [72]. Indeed, reduced platelet serotonin uptake and diminished duodenal SERT mRNA expression have been documented in IBS-D, changes associated with local immune activation characterized by intraepithelial lymphocyte infiltration, mast-cell expansion, and increased tryptase release [73].

Genetic factors may also shape serotonin metabolism. Variants in SERT, 5-HT receptors, and tryptophan hydroxylase (TPH), the enzyme catalyzing serotonin biosynthesis, have been explored with conflicting results. A meta-analysis of the 5-HTTLPR polymorphism involving over 6800 participants found no overall association with IBS, although the LL genotype appeared linked to constipation-predominant IBS in East Asian cohorts [74]. In contrast, a large multi-center study identified the HTR3E variant rs56109847 as a genetic risk factor for female IBS-D and demonstrated reduced HTR3E expression in the sigmoid colon of affected patients [75]. Another study demonstrated that the SLC6A4 rs4795541 and HTR2A rs6311 polymorphisms are linked to depressive disorders in IBS patients, while the HTR2C rs6318 variant appears to influence susceptibility to anxiety disorders in this population [76]. Beyond genetics, immune mediators also regulate serotonin signaling. For instance, IFN-γ has been shown to downregulate SERT expression in vitro. Moreover, colonic biopsies from IBS patients exhibit elevated IFN-γ signaling together with increased numbers of serotonin-positive enterochromaffin cells, particularly in IBS-D [77,78]. These changes correlate with mast-cell density and abdominal pain severity, highlighting the crosstalk between mucosal immunity and serotonin metabolism.

Finally, alterations in serotonin metabolism may also influence barrier integrity. Experimental data have shown that oral serotonin administration increases mucosal 5-HIAA concentrations and decreases expression of the tight junction protein occludin, suggesting that excess luminal serotonin could compromise epithelial integrity [79]. Taken together, these findings highlight serotonin as a central mediator linking luminal sensing, motility regulation, immune activation, and barrier function, thereby shaping the symptom profile of IBS. Beyond serotonin, other neurotransmitters, including dopamine and GABA, further modulate gut function and pain sensitivity, further complicating the pathophysiological landscape of IBS [80].

Additionally, bile acid malabsorption has also been identified as a contributing factor in IBS-D, as elevated bile acids in the colon can disrupt motility and exacerbate diarrhea symptoms [81], highlighting the diverse mechanisms underlying IBS symptomatology. Under normal conditions, bile acids are synthesized in the liver, secreted into the duodenum, and efficiently reabsorbed in the terminal ileum via the apical ileal bile acid transporter [82]. More than 95% are recycled to hepatocytes through the enterohepatic circulation, a process tightly regulated by fibroblast growth factor (FGF)-19, which is induced by activation of the nuclear farnesoid X receptor in enterocytes. FGF-19 then provides negative feedback on hepatic bile acid synthesis via the FGF-4 receptor and the cofactor klotho-β [83,84,85]. When this regulatory circuit is disrupted, excessive bile acids may cross over into the colon, where they increase motility and permeability, and amplify visceral sensitivity. Idiopathic bile acid diarrhea is present in up to 20% of patients with IBS-D [86]. Elevated markers of bile acid synthesis, including 7α-hydroxy-4-cholesten-3-one (C4), have been shown to correlate with increased fecal bile acid concentrations and greater colonic permeability [87,88,89]. Genetic variants in klotho-β and FGF-4 further support a mechanistic link between altered bile acid metabolism and colonic transit abnormalities in IBS-D [87,90].

Moreover, microbial composition influences bile acid levels and their biological effects. Dysbiosis in IBS-D has been associated with higher stool bile acid concentrations, accompanied by increased *Escherichia coli* and reduced *Leptum* and *Bifidobacterium* spp. [91]. Differences in both fecal and serum bile acid profiles have been observed between IBS-D, IBS-C, and healthy individuals, suggesting that the intestinal microbiota actively modulates bile acid metabolism and, consequently, symptom expression [92]. Collectively, these findings highlight that serotonin dysregulation and bile acid imbalance represent shared pathways in IBS, each capable of reshaping motility, permeability, and sensory signaling, and together offering mechanistic targets for future therapies.

## 8. Genetic and Epigenetic Contributions in IBS Pathophysiology: Mechanistic Insights and Clinical Implications

Genetic predispositions also warrant further investigation, as over 60 candidate genes have been associated with IBS [93,94]. These genes are involved in serotonin metabolism, immune system activation, and neuropeptide pathways, reinforcing the complex genetic mechanisms of the disorder. Large-scale genome-wide association studies have identified multiple susceptibility loci for IBS, highlighting shared genetic pathways with mood and anxiety disorders and supporting the role of genetics in impaired brain–gut interactions [95]. In addition, genome-wide studies have revealed that IBS patients bear distinct genetic variants, particularly at the chromosome 9q31.2 locus (rs10512344), which appears to exert female-specific effects [96]. This locus has been mainly related to the regulation of transmembrane ion transport, mutations in the sucrase-isomaltase gene, which code for disaccharidases with deficient enzymatic activity, and impaired function of the autonomic nervous system [96,97,98]. Hypomorphic variants of the sucrase-isomaltase gene have been associated not only with rare recessive forms of sucrose intolerance but also with an increased risk and severity of IBS, as well as differential responses to carbohydrate-reducing diets, thereby supporting the rationale for personalized dietary approaches in affected patients [99]. Additional susceptibility variants for IBS have been identified in the sodium voltage-gated channel alpha subunit 5A (SCN5A) gene [100]. This gene encodes sodium channels that are selectively expressed in the gastrointestinal tract and play a pivotal role in modulating the activity of smooth muscle cells and interstitial cells of Cajal [101]. Beyond ion transport mechanisms, genetic polymorphisms affecting neurotransmission have also been implicated. These include variants in genes responsible for serotonin biosynthesis, such as those encoding the TPH isoforms 1 and 2, as well as in genes regulating serotonin reuptake, notably the SERT [14]. Transcriptomic analysis in IBS has identified 23 candidate genes, with seven putative biomarkers highlighted. Several of these are linked to serotonin metabolism, underscoring the role of altered serotonergic signaling in IBS pathophysiology [102]. In parallel, evidence from experimental models shows that SERT downregulation in IBS-D leads to increased mucosal 5-HT and impaired serotonin clearance, contributing to diarrhea and visceral hypersensitivity via mast cell-mediated pathways [103]. Further GWAS have identified additional risk loci in immune-related genes within the HLA region (such as HLA-C and BAG6) and in neuronal adhesion molecules (NCAM1, CADM2, DOCK9, PHF2), implicating both immune and neuronal pathways in IBS susceptibility [3,104,105,106,107]. These studies also revealed significant overlap with psychiatric traits, including anxiety, depression, and insomnia, consistent with the observed clinical comorbidity and the therapeutic benefit of psychotropic and behavioral interventions [95,107]. Importantly, stool frequency and consistency have been used as quantitative endophenotypes to underscore susceptibility genes such as BDNF, CALCA, CRHR1, and the female-specific FFAR3, all linked to gut motility regulation [96]. Polygenic risk scores derived from these studies not only predict IBS risk but also distinguish between IBS-D and IBS-C, suggesting that opposite directions of the same biological pathways underlie diarrhea versus constipation [3]. Finally, gene-microbiome interactions may also modulate disease expression, with variants in LCT (lactase persistence), FUT2 (secretor status), and ABO influencing microbial composition and dietary tolerance [108,109], thus offering additional avenues for personalized interventions, although none of these three loci has been directly associated with IBS so far.

Epigenetic alterations have emerged as important contributors to the pathophysiology of IBS as well, particularly in the context of brain–gut axis dysregulation. These regulatory processes involve chromosome-dependent heritable modifications in gene expression without altering the underlying DNA sequence [110]. Beyond DNA methylation and histone modifications, accumulating evidence highlights the role of early-life stress in inducing lasting epigenetic reprogramming of HPA axis-related genes, such as NR3C1 and CRF, thereby sensitizing individuals to visceral pain and stress-related symptom exacerbation [111,112]. Sex-specific differences in IBS prevalence may also be partially explained by estrogen-dependent epigenetic regulation of serotonin pathways and pain perception networks [113,114].

In this setting, disturbances in the gut microbiota may modulate the production of metabolites such as sodium butyrate, a well-recognized inhibitor of histone deacetylases [115]. Although, many studies have demonstrated a reduction in butyrate-producing microbial phyla in IBS, especially IBS-D and IBS-M subtypes [116], along with distinct alterations in gene methylation and microRNA (miRNA) expression [110], other microbial metabolites, including propionate, acetate, and tryptophan derivatives, may also influence histone acetylation and methylation, thereby modulating barrier function, motility, and immune tolerance [117]. Importantly, this relationship is bidirectional; extracellular miRNAs released by intestinal epithelial and stem cells help maintain microbial equilibrium [118,119], while microbial signals and metabolites reciprocally shape epigenetic landscapes and host gene expression [119,120]. Intestinal epithelial cells, including crypt enterocytes, are emerging as key mediators of host–microbiome interaction, regulated through both miRNA signaling [118] and histone modifications [121].

Members of the miR-199 family have been implicated in visceral hypersensitivity and increased intestinal permeability and thus may represent promising biomarkers for IBS diagnosis and treatment, especially in IBS-D [122,123]. Beyond post-transcriptional regulation, IBS patients also demonstrate DNA methylation in regulatory genes. These include genes linked to oxidative stress responses, such as glutathione-S-transferase Mu 5, as well as genes regulating the HPA axis, most notably the CRF gene [14,110].

Collectively, these findings suggest that epigenetic modifications may mediate the interaction between environmental triggers, such as diet, stress, or infection, and genetic susceptibility, ultimately shaping IBS phenotypes through modified gene expression. Importantly, as epigenetic alterations are potentially reversible, they represent promising targets for novel therapeutic strategies, ranging from dietary interventions and psychobiotics to epigenetic drugs and miRNA-based approaches. Nevertheless, further large-scale and longitudinal studies are needed to validate these mechanisms as reliable biomarkers and to translate them into clinical practice.

Overall, genetic, epigenetic, and microbial mechanisms form an interdependent network that shapes IBS heterogeneity. The presence of genetic polymorphisms in serotonergic, immune, and barrier-regulating genes may predispose individuals to impaired intestinal physiology, while epigenetic modifications act as mediators translating environmental influences, such as diet, stress, or infection, into transcriptional changes. The gut microbiome represents both a target and a driver of these processes, influencing host gene expression through microbial metabolites and, in turn, being shaped by host genetic background and epigenetic reprogramming. This bidirectional interplay creates a feedback loop through which genetic susceptibility, epigenetic plasticity, and microbial composition modulate symptoms, severity, and therapeutic responses. Understanding this convergence provides a step toward integrative biomarkers and personalized therapeutic approaches in IBS. By integrating these molecular mechanisms, the gut microbiome emerges as a unifying factor that connects neural, immune, and metabolic pathways in IBS.

## 9. Gut Microbiome and Metabolites: Integrating Microbial Functions with Clinical Outcomes

The gut microbiota plays a central role in this neuroimmune–gastrointestinal network, influencing neurotransmitter synthesis, including serotonin, GABA, and dopamine, through microbial metabolites such as short-chain fatty acids (SCFAs), tryptophan catabolites, and bile acid derivatives [124,125]. SCFAs, mainly butyrate, can modulate enteric nerve cell responsiveness and glial cell activation, thereby influencing motility and visceral sensitivity [15,126,127]. These insights align with recent findings demonstrating that targeted manipulation of the gut microbiota can beneficially influence mucosal immunity, epithelial repair, and gut–brain communication in inflammatory and functional gastrointestinal disorders [128]. Importantly, clinical evidence supports these mechanistic links. In a randomized trial of fecal microbiota transplantation in IBS, fecal butyric acid levels significantly increased after donor-FMT. Moreover, higher butyric acid concentrations were inversely correlated with symptom severity and fatigue scores, whereas no such changes were observed in the placebo group [NCT03822299] [129]. These findings reinforce the concept that SCFAs, particularly butyrate, are not only critical mediators of mucosal and neuronal function but also are directly linked to symptomatic improvement in IBS. Dysbiosis, particularly a disrupted Firmicutes-to-Bacteroidetes ratio, leads to reduced SCFA production and altered microbial-derived indole, which in turn affects epithelial integrity and mucosal immunity [130]. This microbial imbalance facilitates low-grade inflammation and enhances gut permeability, promoting the translocation of bacterial components that further sensitize the ENS [15]. More specifically, certain taxa directly influence gut neurotransmission and the serotonergic system. For instance, *Bifidobacterium dentium* increases intestinal serotonin levels, upregulates serotonin receptor expression, and produces GABA [131,132]. Similarly, *B. adolescentis* also produces GABA, linking luminal metabolism to the brain–gut axis [132].

However, the role of SCFAs, particularly butyrate, in IBS pathophysiology remains complex and, to some extent, controversial. As previously discussed, most studies highlight their anti-inflammatory and barrier-protective effects. However, there is evidence that excessive luminal butyrate may paradoxically enhance visceral pain signaling or exacerbate motility disturbances in susceptible individuals. Experimental studies have shown that sodium butyrate can induce colonic hypersensitivity in rats through enteric glial cell-mediated upregulation of nerve growth factor (NGF), linking butyrate exposure to enhanced nociceptive signaling [133]. Additional data have reported that butyrate can stimulate enteric neurons and promote colonic contractions at high concentrations, whereas lower doses may exert antinociceptive or anti-inflammatory effects [134]. Similarly, clinical data on fecal SCFA levels in IBS patients are inconsistent, with some studies reporting elevated concentrations in certain subtypes, whereas others report reduced or unchanged levels for acetate, butyrate or total SCFAs, findings that likely reflect heterogeneity in IBS subtypes, diet, transit time and analytical methods [135]. These discrepancies may also result from variations in study design, microbial diversity, and individual metabolic responses. Therefore, while SCFAs remain promising modulators of gut physiology, their contradictory effects highlight the need for individualized approaches when targeting microbial metabolism in IBS.

Taken together, these findings underscore that alterations in microbial metabolites can disrupt epithelial integrity and sensory signaling, thereby setting the stage for downstream immune activation pathways, including those mediated by bacterial components. Lipopolysaccharide activates TLR4 on mast cells, stimulating histamine, tryptase, and prostaglandin release, which disrupts tight junctions and enhances visceral hypersensitivity [42]. Barrier dysfunction is further aggravated by fecal proteases, which are elevated in a significant proportion of post-infectious IBS patients, that degrade epithelial junctional proteins [136,137]. Beyond their direct epithelial effects, fecal proteases may also influence host–microbe interactions via FK506 binding protein (FKBP)-type peptidylprolyl cis-trans isomerases, which regulate immune responses and glucocorticoid receptor activity [138]. In this context, a randomized trial showed that treatment with the serine protease inhibitor camostat mesilate (CM) in IBS patients modulated the fecal microbiome, increasing the abundance of *Streptococcus* and enhancing functional pathways related to serine protease and FKBP-type PPIases. However, symptomatic relief did not differ from that observed in the placebo group [138].

In parallel, mucin-degrading bacteria such as *Ruminococcus gnavus*, *R. torques*, and *Akkermansia muciniphila* influence the production of the protective mucus layer by reducing the thickness [139,140]. Microbial metabolites, including SCFAs, indoles, and secondary bile acids, modulate motility and secretion by stimulating enteroendocrine release of peptide YY, glucagon-like peptide-1 (GLP-1), and serotonin [141,142,143], while tryptamine derived from tryptophan catabolism activates 5-HT4 receptors to enhance gut motility reflexes [23]. In contrast, methane produced by *Methanobrevibacter smithii* decreases intestinal transit and is strongly associated with constipation-predominant IBS [144], whereas hydrogen sulfide generated by sulfate-reducing bacteria impairs smooth muscle contractility [145]. Diet strongly modulates these interactions, as high-FODMAP intake enriches Gram-negative bacteria and increases luminal LPS, thereby activating LPS-TLR4 signaling pathways, worsening barrier integrity, and increasing colonic sensitivity, whereas low-FODMAP diets can partially reverse these effects [146]. Similarly, pathogens including *Campylobacter*, *Salmonella*, or *Giardia* lead to higher counts of enterochromaffin cells, mast cells, and mucosal immune activation, resulting in a pro-inflammatory milieu that predisposes to chronic hypersensitivity [147,148,149,150]. Collectively, these pathways illustrate how dysbiosis modulates epithelial and immune homeostasis, enhances neuronal responsiveness in the ENS, and disrupts neuromodulatory balance, ultimately perpetuating visceral hypersensitivity and intestinal motility disturbances in IBS.

Despite promising mechanistic insights, clinical translation of microbiome modulation in IBS remains inconsistent. Variability in donor profiles, host genetics, microbial resilience, and diverse study designs contribute to heterogeneous outcomes with FMT and probiotics. These findings highlight the need for standardized methodologies and consideration of host–microbe interactions to enable reproducible and personalized microbiome-based interventions.

Table 1 provides an overview of the major molecular, microbial, metabolic, and genetic mechanisms linked to IBS-D and IBS-C subtypes.

## 10. Conclusions

The pathophysiology of IBS is undoubtedly complex, encompassing diverse mechanisms that extend beyond the gastrointestinal tract, including dysregulated neurotransmitter systems, impaired immune responses and gut microbiota imbalance (Figure 1).

Figure 1 illustrates the key pathophysiological mechanisms underlying irritable bowel syndrome (IBS), focusing on the interplay between the brain–gut axis (BGA), enteric nervous system (ENS), immune system activation, gut microbiota and environmental factors. The hypothalamus, through the hypothalamic–pituitary–adrenal (HPA) axis, releases the corticotropin-releasing hormone (CRH), which stimulates the pituitary gland to secrete the adrenocorticotropic hormone (ACTH). This, in turn, induces the adrenal cortex to release cortisol, a stress-related hormone that influences gut function and pain perception through the central nervous system (CNS). CNS and ENS communicate bidirectionally via the vagus nerve, affecting pain perception, gut motility, and visceral sensitivity. Serotonin (5-HT), released by enterochromaffin cells in the gut, plays a crucial role in regulating gut motility and peristalsis. Impaired regulation in serotonin metabolism contributes to IBS symptoms such as diarrhea and constipation. Dietary factors, including fermentable oligosaccharides, disaccharides, monosaccharides, polyols (FODMAPs), and dietary fiber, influence gut microbiota composition and the production of short-chain fatty acids (SCFAs). These SCFAs impact gut barrier integrity and immune modulation. Alterations in bile acid synthesis impact gut motility and intestinal barrier function. Elevated intestinal permeability, mediated by impaired tight junctions (TJs), contributes to visceral hypersensitivity and low-grade inflammation. The gut microbiota, through microbial metabolites, interacts with the immune system, influencing gut barrier function and inflammation. The activation of the immune system, which involves macrophages, dendritic cells (DCs) and mast cells, releases pro-inflammatory cytokines, histamine, and proteases, further contributing to visceral hypersensitivity, impaired barrier function and pain perception. Genetic predisposition also plays a role, affecting serotonin metabolism, immune activation, and neuropeptide pathways. This proposed model underscores the multifaceted nature of IBS, emphasizing the role of neuroimmune interactions, gut microbiota, and environmental influences in disease pathogenesis. This figure was created with BioRender.com (accessed on 20 September 2025). Abbreviations: HPA, hypothalamic–pituitary–adrenal; CRH, corticotropin-releasing hormone; ACTH, adrenocorticotropic hormone; CNS, central nervous system; ENS, enteric nervous system; BGA, brain–gut axis; SCFAs, short-chain fatty acids; 5-HT, serotonin; DC, dendritic cell; TJ, tight junction.

Evidence discussed in this review emphasizes the critical role of the brain–gut axis, where dysregulation of neuroendocrine signaling, particularly HPA hyperactivity, exacerbates intestinal motility, visceral hypersensitivity, and stress-related symptom progression. In parallel, disturbances in serotonin signaling and bile acid metabolism emerge as critical drivers of subtype-specific manifestations, such as diarrhea- or constipation-predominant IBS.

Immune activation and impaired intestinal barrier function further enhance this process. Low-grade mucosal inflammation, mast-cell activation, and cytokine imbalance contribute to pain perception and impaired permeability, while stress-induced cortisol release perpetuates immune dysregulation. Genomic studies implicate multiple susceptibility variants in neurotransmission, ion transport, and immune regulation, whereas epigenetic mechanisms, including microRNA signaling and DNA methylation, reveal how environmental triggers shape IBS phenotypes.

The gut microbiome represents another substantial pathogenic factor, influencing serotonin and GABA metabolism, bile acid composition, and mucosal integrity. Dysbiosis, often characterized by the reduced production of SCFAs and increased pro-inflammatory metabolites, directly links microbial imbalance with barrier disruption and neuronal sensitization. Nutritional interventions, including low-FODMAP diets, further underscore the importance of diet–microbiome interactions in symptom modulation.

Taken together, these insights demonstrate that IBS cannot be regarded as an exclusively gastrointestinal disorder but rather as a systemic condition arising from intricate crosstalk among neural, immune, microbial, genetic, and environmental complex interactions. This complexity underscores the need for multidimensional management strategies that address not only gastrointestinal symptoms but also psychological and neurobiological dimensions. A holistic framework that integrates genetic susceptibility, epigenetic regulation, and microbiome–host interactions is essential for delineating IBS phenotypic diversity and guiding the development of personalized therapeutic strategies.

However, this review has limitations. There is evidence, which is derived from observational or cross-sectional studies, limiting causal inference. In addition, several proposed mechanisms are primarily supported by animal models, and their direct clinical relevance requires validation in human studies. The considerable heterogeneity among study populations, diagnostic criteria, and analytical methodologies further complicates the interpretation of findings. Addressing these limitations through standardized, large-scale, and longitudinal human studies will be essential in strengthening mechanistic understanding and clinical translation.

Despite these challenges, several research gaps remain to be addressed. There is a lack of long-term cohort studies tracking microbial evolution and its relationship with disease progression and symptom variability. Moreover, mechanistic studies integrating multi-omics data are needed to clarify causality. From a translational point of view, the development of microbiome-based diagnostic kits and predictive markers may facilitate clinical stratification and personalized treatment strategies. Finally, future intervention trials should focus on the design of clinical studies that combine psychological, microbial, and dietary approaches, targeting the brain–gut–microbiota axis.

Looking ahead, advances in genomics, microbiome modulation, and stress regulation offer opportunities for personalized therapies. Integration of host–microbe interactions, dietary influences, and genetic risk stratification may ultimately pave the way toward precision medicine in IBS. Future research should prioritize longitudinal and integrative studies that bring together these systems, identify reliable biomarkers, and translate mechanistic insights into tailored interventions capable of improving both symptom control and quality of life for individuals living with this challenging disorder.

## Figures and Tables

**Figure 1 ijms-26-10598-f001:**
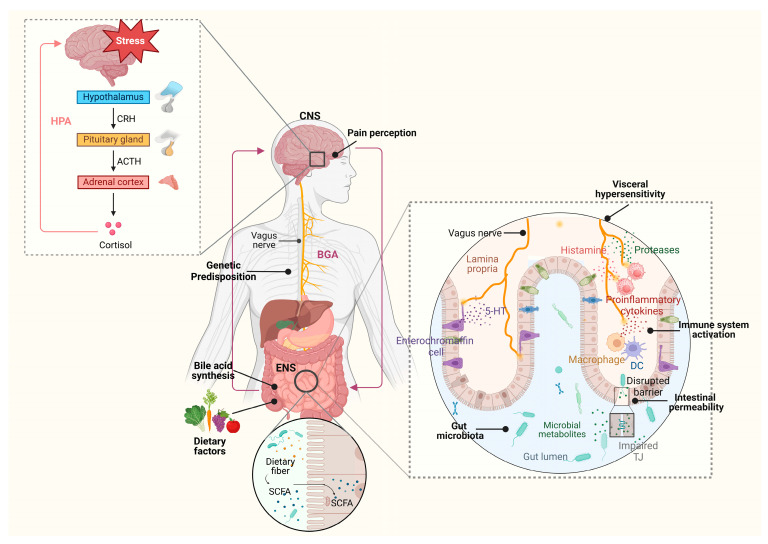
Pathophysiological mechanisms of irritable bowel syndrome (IBS): a multifactorial overview.

**Table 1 ijms-26-10598-t001:** Major molecular and microbial mechanisms associated with IBS subtypes.

Molecular Mechanism Involved	IBS-D (Diarrhea-Predominant)	IBS-C (Constipation-Predominant)	Not Subtype-Specific
Serotonin signaling	↑ Postprandial 5-HT release; ↓ Platelet 5-HT uptake; ↓ Duodenal SERT mRNA; ↑ Enterochromaffin cells; HTR3E rs56109847 variant; IFN-γ–induced ↓ SERT	↓ Postprandial 5-HT release; 5-HTTLPR LL genotype linked to IBS-C (East Asian cohorts)	NR
Bile acid metabolism	↑ Colonic bile acids (impaired FGF-19 feedback); FGF-4 and klotho-β variants; idiopathic bile acid diarrhea (~20%)	NR	NR
Immune activation	↑ Mast cells, eosinophils, lymphocytes; ↑ IL-6, IL-8, TNFα; ↑ Mucosal B cells and IgA-coated bacteria; ↑ CRF in mucosal eosinophils	Mild mucosal immune activation	↓ β-Endorphin release from immune cells
Barrier integrity	↑ Intestinal permeability; ↓ ZO-1, occludin, claudin-1; ↑ zonulin; TESK1/CFL downregulation	↑ Intestinal permeability (4–25%); ↑ Mast-cell activation; ↑ TRPV1 fibers	NR
Microbial alterations	↑ *E. coli*; ↓ *Leptum*, *Bifidobacterium* spp.	↑ *Methanobrevibacter smithii* (methane producers)	↑ Fecal bile acids; ↑ Fecal proteases; disrupted Firmicutes/Bacteroidetes ratio; altered SCFAs/indoles; impaired epithelial integrity
Metabolites and neurotransmitters	↑ Butyrate, tryptamine; ↑ motility (dose-dependent effects)	↑ Methane → slowed transit	Altered SCFAs, tryptamine, indoles, and GABA signaling
Genetic/epigenetic influences	HTR3E rs56109847; SERT variants; IFN-γ–induced ↓ SERT expression; ↓ miR-199a/b expression → ↑ visceral hypersensitivity	5-HTTLPR LL genotype; SCN5A polymorphisms	Immune-related gene variants within the HLA region

Symbols: ↑ Increased/Upregulated; ↓ Decreased/Downregulated; → Results in. Abbreviations: NR, Not available or not explicitly reported; 5-HT, Serotonin (5-hydroxytryptamine); 5-HIAA, 5-Hydroxyindoleacetic acid; SERT→, Serotonin transporter; HTR3E, 5-Hydroxytryptamine (serotonin) receptor 3E subunit; 5-HTTLPR, Serotonin-transporter-linked polymorphic region; IFN-γ, Interferon gamma; miR-199a/b, MicroRNA-199a and MicroRNA-199b; SCN5A, Sodium voltage-gated channel alpha subunit 5 gene; HLA, Human leukocyte antigen; CRF, Corticotropin-releasing factor; TESK1, Testis-specific kinase 1; CFL1, Cofilin 1; ZO-1, Zonula occludens-1; TRPV1, Transient receptor potential vanilloid type 1; NGF, Nerve growth factor; FGF-19, Fibroblast growth factor 19; SCFAs, Short-chain fatty acids; GABA; Gamma-aminobutyric acid.

## Data Availability

No new data were created or analyzed in this study.

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
