# Peer review of "Unraveling the Pathophysiology of Irritable Bowel Syndrome: Mechanisms and Insights"

_ijms, 2025, doi:10.3390/ijms262110598_

Round 1
Reviewer 1 Report
Comments and Suggestions for Authors
The review article provides a detailed overview of the molecular, microbial, and neuroimmune mechanisms involved in Irritable Bowel Syndrome (IBS), particularly focusing on genetic, epigenetic, and microbiota-mediated pathways. The topic is highly relevant and well aligned with the journal’s scope. However, several quiries should be addressed to improve the clarity, scientific and overall quality of the manuscript.
1-The manuscript discusses genetic, epigenetic, and microbial factors separately but does not adequately integrate how these pathways converge to modulate IBS phenotypes.
2-The manuscript discusses genetic, epigenetic, and microbial factors separately but does not adequately integrate how these pathways converge to modulate IBS phenotypes.
3-Section-8 relies heavily on summarizing existing studies without providing critical analysis or highlighting controversies. For instance, while the role of SCFAs, especially butyrate, is discussed, the opposing evidence regarding their variable effects on motility and inflammation in IBS patients is not mentioned. The authors should provide a more balanced discussion
4-occasionally shifts between clinical and molecular perspectives without clear transitions. Subheadings could be refined to improve logical flow
5-There are some grammatical inconsistencies (e.g., “increased abundance of butyrate-producing microbial phyla” should specify whether it refers to IBS-D or IBS-C patients).
6-Author should add a concise table summarizing major molecular and microbial mechanisms associated with each IBS subtype (IBS-D, IBS-C, IBS-M).
Author Response
Reviewer 1
The review article provides a detailed overview of the molecular, microbial, and neuroimmune mechanisms involved in Irritable Bowel Syndrome (IBS), particularly focusing on genetic, epigenetic, and microbiota-mediated pathways. The topic is highly relevant and well aligned with the journal’s scope. However, several queries should be addressed to improve the clarity, scientific and overall quality of the manuscript.
Response to Reviewer 1: We thank reviewer 1 for his positive feedback on our manuscript.
Comment 1: The manuscript discusses genetic, epigenetic, and microbial factors separately but does not adequately integrate how these pathways converge to modulate IBS phenotypes.
Response to comment 1: We thank the reviewer for this insightful comment. We have now elaborated on Section 8 in the revised manuscript to include a new integrative paragraph that discusses how genetic, epigenetic, and microbial mechanisms interact to shape IBS heterogeneity and symptom expression (pages 8-9, lines 379-392). In addition, we have added a short statement in the Conclusion highlighting the importance of a holistic framework integrating these pathways (page 14, lines 551-553).
Comment 2: Section-8 relies heavily on summarizing existing studies without providing critical analysis or highlighting controversies. For instance, while the role of SCFAs, especially butyrate, is discussed, the opposing evidence regarding their variable effects on motility and inflammation in IBS patients is not mentioned. The authors should provide a more balanced discussion.
Response to comment 2: We thank the reviewer for highlighting this important discrepancy in the role of SCFAs. We have now revised this section to include a critical discussion of the conflicting evidence on SCFAs, particularly butyrate, emphasizing their context-dependent effects on motility, inflammation, and visceral sensitivity (pages 9-10, lines 420-437).
Comment 3: Occasionally shifts between clinical and molecular perspectives without clear transitions. Subheadings could be refined to improve logical flow.
Response to comment 3: To enhance the coherence between molecular mechanisms and clinical aspects, we have revised several subheadings to clarify the focus of each section and added brief transition sentences to improve narrative flow (page 3, lines 129-131 and 134-136, page 5, lines 204-209, page 8, lines 438-440).
Comment 4: There are some grammatical inconsistencies (e.g., “increased abundance of butyrate-producing microbial phyla” should specify whether it refers to IBS-D or IBS-C patients).
Response to comment 4: We thank the Reviewer for this comment. Indeed, there were some inconsistencies in the original paragraph regarding the abundance of butyrate-producing microbial taxa in IBS. We have carefully reviewed the relevant literature and revised the text accordingly to reflect the current evidence (page 8, lines 350-364).
Comment 5: Author should add a concise table summarizing major molecular and microbial mechanisms associated with each IBS subtype (IBS-D, IBS-C, IBS-M).
Response to comment 5: We thank the reviewer for this suggestion. In the revised manuscript, we have added Table 1, which provides a concise summary of the major molecular, microbial, metabolic, and genetic mechanisms associated with IBS-D and IBS-C subtypes. The IBS-M subtype was not included, as data on this group were sparse within the current manuscript (pages 10-11, lines 478-490).
Reviewer 2 Report
Comments and Suggestions for Authors
This is a comprehensive and well-structured review that successfully integrates current mechanistic insights into the multifactorial pathophysiology of irritable bowel syndrome (IBS). The narrative is coherent, logically organized, and supported by relevant studies.
Decision: Minor Revision.
Introduction
- The introduction provides solid background but can be improved by clarifying the evolving classification of IBS within the broader context of disorders of gut-brain interaction (DGBI).
- You could add a short paragraph linking early-life stress, microbiota development, and brain function, which would help bridge gut-brain communication with developmental and psychological components of IBS.
- After line 60–63, where microbiota’s influence on gene expression and host responses is discussed, I suggest to support this paragraph with updated references such as https://doi.org/10.1007/s12035-025-04846-0 , This recent review provides an updated understanding of how gut microbial metabolites regulate neurotransmitter systems, stress responses, and mental health, directly supporting the section discussing CNS and microbiota interplay.
Within Section 8
- (lines ~344–350), where the text reads: “SCFAs, mainly butyrate, can modulate enteric nerve cell responsiveness and glial cell activation, thereby influencing motility and visceral sensitivity...” I suggest adding: “These insights align with recent findings demonstrating that targeted manipulation of the gut microbiota can beneficially influence mucosal immunity, epithelial repair, and gut–brain communication in inflammatory and functional gastrointestinal disorders https://doi.org/10.3390/immuno4040026
- Consider adding a critical appraisal of why clinical translation of microbiome modulation (e.g., FMT, probiotics) remains inconsistent—highlighting heterogeneity, donor effects, and host-genetic interactions.
Author Response
Reviewer 2
This is a comprehensive and well-structured review that successfully integrates current mechanistic insights into the multifactorial pathophysiology of irritable bowel syndrome (IBS). The narrative is coherent, logically organized, and supported by relevant studies.
Response to Reviewer 1: Thank you very much for your positive feedback on our manuscript.
Comment 1: The introduction provides solid background but can be improved by clarifying the evolving classification of IBS within the broader context of disorders of gut-brain interaction (DGBI). You could add a short paragraph linking early-life stress, microbiota development, and brain function, which would help bridge gut-brain communication with developmental and psychological components of IBS. After line 60–63, where microbiota’s influence on gene expression and host responses is discussed, I suggest to support this paragraph with updated references such as https://doi.org/10.1007/s12035-025-04846-0 , This recent review provides an updated understanding of how gut microbial metabolites regulate neurotransmitter systems, stress responses, and mental health, directly supporting the section discussing CNS and microbiota interplay.
Response to comment 1: We thank the reviewer for this comment. We have revised the Introduction to better clarify the evolving classification of IBS within the framework of disorders of gut-brain interaction (DGBI). In addition, a short paragraph has been added linking early-life stress, microbiota development, and brain function, and how these may predispose to IBS later in life (page 2, lines 62-67). Moreover, we have cited the Review with doi: 10.1007/s12035-025-04846-0, as suggested.
Comment 2: Within Section 8 (lines ~344–350), where the text reads: “SCFAs, mainly butyrate, can modulate enteric nerve cell responsiveness and glial cell activation, thereby influencing motility and visceral sensitivity...” I suggest adding: “These insights align with recent findings demonstrating that targeted manipulation of the gut microbiota can beneficially influence mucosal immunity, epithelial repair, and gut–brain communication in inflammatory and functional gastrointestinal disorders https://doi.org/10.3390/immuno4040026
Response to comment 2: We have revised as suggested (page 9, lines 401-404).
Comment 3: Consider adding a critical appraisal of why clinical translation of microbiome modulation (e.g., FMT, probiotics) remains inconsistent—highlighting heterogeneity, donor effects, and host-genetic interactions.
Response to comment 3: We thank the reviewer for this suggestion. A critical appraisal has been added to the revised manuscript to address why clinical translation of microbiome modulation remains inconsistent (page 10, lines 472-477).
Reviewer 3 Report
Comments and Suggestions for Authors
This article provides a systematic review of the complex pathophysiological mechanisms of Irritable Bowel Syndrome (IBS), emphasizing its nature as a disorder of gut-brain interaction involving interactions among multiple factors. Core mechanisms include dysfunction of the brain-gut axis, such as activation of the hypothalamic-pituitary-adrenal axis and neuroendocrine disturbances; abnormal activation of the intestinal mucosal immune system and low-grade inflammation; impaired intestinal epithelial barrier function and increased permeability; and gut microbiota dysbiosis along with the regulation of neurotransmitters, immunity, and barrier function by microbial metabolites. Furthermore, aberrant serotonin signaling and bile acid metabolism dysregulation play key roles in the symptom-based subtyping of IBS. Genetic predisposition and epigenetic regulation further shape the disease phenotype. The article concludes that IBS is a multisystem disorder whose management requires personalized strategies integrating neurological, immune, microbial, and psychological dimensions. It is deemed worthy of publication. However, the following issues need to be addressed prior to publication, as detailed below:
- The current abstract primarily summarizes the pathological mechanisms of IBS but does not clearly state the unique contribution of this review compared to existing ones. It is recommended to add a statement at the end of the abstract clarifying the novel perspective or structural strengths of this review.
- At the end of the introduction, the scope of the review should be more explicitly defined,for instance, stating whether it covers pediatric IBS, animal models,and the search strategy, such as databases, time range, should be included to enhance methodological transparency.
- The current conclusion is quite general. It is recommended to supplement it with specific research gaps,such as "lack of long-term cohort studies tracking microbial evolution", clinical translation pathways, such as "development of microbiome-based diagnostic kits", and future intervention strategies,such as "design of clinical trials combining psychological, microbial, and dietary interventions".
- It is advisable to add a "Limitations" section before or within the "Conclusion" to outline the limitations of this review. For example: reliance on observational studies limits causal inference; evidence for some mechanisms primarily comes from animal models; high heterogeneity among studies makes unified interpretation challenging.
- To enhance clarity, the introductory content could be appropriately revised. The following are recently published relevant references for consideration:
Journal of Orthopaedic Translation ,doi:10.1016/j.jot.2025.05.008
Small Science,doi:10.1002/smsc.202400474
Journal of Nanobiotechnology,doi : 10.1186/s12951-024-02917-3
Gut Microbes,doi:10.1080/19490976.2023.2295432
Author Response
Reviewer 3
This article provides a systematic review of the complex pathophysiological mechanisms of Irritable Bowel Syndrome (IBS), emphasizing its nature as a disorder of gut-brain interaction involving interactions among multiple factors. Core mechanisms include dysfunction of the brain-gut axis, such as activation of the hypothalamic-pituitary-adrenal axis and neuroendocrine disturbances; abnormal activation of the intestinal mucosal immune system and low-grade inflammation; impaired intestinal epithelial barrier function and increased permeability; and gut microbiota dysbiosis along with the regulation of neurotransmitters, immunity, and barrier function by microbial metabolites. Furthermore, aberrant serotonin signaling and bile acid metabolism dysregulation play key roles in the symptom-based subtyping of IBS. Genetic predisposition and epigenetic regulation further shape the disease phenotype. The article concludes that IBS is a multisystem disorder whose management requires personalized strategies integrating neurological, immune, microbial, and psychological dimensions. It is deemed worthy of publication. However, the following issues need to be addressed prior to publication, as detailed below:
Response to Reviewer 3: Thank you very much for your positive feedback and for taking the time to thoroughly review the manuscript.
Comment 1: The current abstract primarily summarizes the pathological mechanisms of IBS but does not clearly state the unique contribution of this review compared to existing ones. It is recommended to add a statement at the end of the abstract clarifying the novel perspective or structural strengths of this review.
Response to comment 1: We have added a sentence to the abstract section to highlight the novelty of the current manuscript, as suggested (page 1, lines 22-24).
Comment 2: At the end of the introduction, the scope of the review should be more explicitly defined, for instance, stating whether it covers pediatric IBS, animal models, and the search strategy, such as databases, time range, should be included to enhance methodological transparency.
Response to comment 2: We agree with the reviewer’s comment; however, we consider the Aim section not the appropriate place for methodological details. Accordingly, a brief “Materials and Methods” section has been added to describe the literature search, as suggested (page 2, lines 80-88).
Comment 3: The current conclusion is quite general. It is recommended to supplement it with specific research gaps,such as "lack of long-term cohort studies tracking microbial evolution", clinical translation pathways, such as "development of microbiome-based diagnostic kits", and future intervention strategies,such as "design of clinical trials combining psychological, microbial, and dietary interventions".
Response to comment 3: We thank the reviewer for this suggestion. The conclusion has been expanded to include specific research gaps, clinical translation pathways, and future intervention strategies, as suggested (page 14, lines 562-569).
Comment 4: It is advisable to add a "Limitations" section before or within the "Conclusion" to outline the limitations of this review. For example: reliance on observational studies limits causal inference; evidence for some mechanisms primarily comes from animal models; high heterogeneity among studies makes unified interpretation challenging.
Response to comment 4: A paragraph outlining the main limitations of this review has been added within the Conclusion section, as suggested (page 14, lines 554-561).
Comment 5: To enhance clarity, the introductory content could be appropriately revised. The following are recently published relevant references for consideration: Journal of Orthopaedic Translation , doi:10.1016/j.jot.2025.05.008; Small Science,doi:10.1002/smsc.202400474; Journal of Nanobiotechnology,doi : 10.1186/s12951-024-02917-3; Gut Microbes ,doi:10.1080/19490976.2023.2295432.
Response to comment 5: Thank you for providing the following recent articles:
- DOI 10.1016/j.jot.2025.05.008 (Journal of Orthopaedic Translation)
- DOI 10.1002/smsc.202400474 (Small Science)
- DOI 10.1186/s12951-024-02917-3 (Journal of Nanobiotechnology)
- DOI 10.1080/19490976.2023.2295432 (Gut Microbes)
While we appreciate the suggestion and have reviewed the articles, we have determined that these references are not sufficiently aligned with the specific mechanistic focus and clinical translation pathway of our manuscript on the neuroimmune, microbial, and molecular mechanisms underlying irritable bowel syndrome. Although these studies are indeed novel and interesting, we therefore opted not to include them in the revised reference list, as not strongly related to IBS, in order to maintain thematic coherence and avoid loss of study focus.
Quality of English Language: The English could be improved to more clearly express the research.
We thank the reviewer for this comment. The entire manuscript has been thoroughly reviewed, and all suggested changes have been incorporated where applicable. Moreover, the revised version has been edited by a professional English language editor to improve clarity and overall readability.